# Genetic mapping of cell type specificity for complex traits

Kyoko Watanabe [1], Maša Umićević Mirkov[1], Christiaan A. de Leeuw [1], Martijn P. van den Heuvel [1,2] & Danielle Posthuma [1,2]

Single-cell RNA sequencing (scRNA-seq) data allows to create cell type specific transcriptome profiles. Such profiles can be aligned with genome-wide association studies (GWASs) to implicate cell type specificity of the traits. Current methods typically rely only on a small subset of available scRNA-seq datasets, and integrating multiple datasets is hampered by complex batch effects. Here we collated 43 publicly available scRNA-seq datasets. We propose a 3-step workflow with conditional analyses within and between datasets, circumventing batch effects, to uncover associations of traits with cell types. Applying this method to 26 traits, we identify independent associations of multiple cell types. These results lead to starting points for follow-up functional studies aimed at gaining a mechanistic understanding of these traits. The proposed framework as well as the curated scRNA-seq datasets are made available via an online platform, FUMA, to facilitate rapid evaluation of cell type specificity by other researchers.

[1] Department of Complex Trait Genetics, Center for Neurogenomics and Cognitive Research, Neuroscience Campus Amsterdam, VU University Amsterdam, Amsterdam, The Netherlands. [2] Department of Clinical Genetics, section Complex Trait Genetics, Neuroscience Campus Amsterdam, VU Medical Center, Amsterdam, The Netherlands. Correspondence and requests for materials should be addressed to D.P. (email: d.posthuma@vu.nl)

More than a decade of genome-wide association studies (GWASs) have yielded genetic risk variants for a wide variety of traits including psychiatric disorders, neuro-degenerative, cardiovascular, and metabolic disease, as well as quantitative traits, such as intelligence (IQ), educational attainment (EA), and height[1]. The emerging picture is that complex traits are generally highly polygenic, with hundreds or even thousands of risk variants each contributing a small incremental effect. Although the identification of such risk variants is a major step forward in our understanding of disease etiology and individual differences in human traits, the high polygenicity of traits also poses a challenge. Gaining biological insight from GWASs entails considering the polygenic nature of traits, and rethinking what we see as starting points for follow-up functional work.

Several recent studies have identified the involvement of specific tissues in complex traits by integrating tissue-specific gene expression profiles with GWAS summary statistics[2–6]. Multiple traits indeed showed significant enrichment in relevant tissue types, such as schizophrenia (SCZ) and body mass index (BMI) in brain[7,8], waist hip ratio adjusted for BMI (WHR) in adipose tissue[9] and pulse pressure in the heart and arteries[10]. However, human primary tissue samples are often a mixture of multiple cell types, and the expression level of a gene in a certain tissue is an average of the cell type-specific expression weighted by the proportion of the cell types in the tissue samples. Tissue-specific expression is thus a function of the distribution of cell types present in that tissue. In addition, tissue-specific expression levels can also be influenced by characteristics of the sampling process, such as the specific region of the tissue that is sampled, which may influence the distribution of cell types. Considering the heterogenous population of cell types in a tissue, investigating the genetically mediated association of specific cell types with traits may provide more specific information, which is important for gaining mechanistic insights. Indicating the cell types that are involved in complex traits can guide designing functional follow-up experiments. For example, recent Cre-mediated DREADD technology allows to investigate effects of controlling the function of a particular cell type[11,12]. We thus argue that the prioritization of specific cell types needs to be part of the standard post-GWAS annotation pipeline, and its analysis needs to be readily available to researchers in the GWAS field.

In recent years, single-cell RNA sequencing (scRNA-seq) technology has dramatically improved. Over a thousand cells can now be sequenced with high quality in a single study, and several studies have made their datasets publicly available[13–15]. Compared to the traditional bulk RNA-seq, scRNA-seq can characterize transcriptome profiles (TPMs) of specific cell types in much higher resolution, as is done for example in large-scale scRNA-seq studies such as the Human Cell Atlas[16], Mouse Cell Atlas[17], and Tabula Muris[18]. Recent studies linking cell-specific expression profiles with GWASs, have already implicated cell types involved in, e.g. SCZ[5,19], IQ[20], and neuroticism (NEU)[21]. While progress is made, these studies are still in their infancy; typically, only a few scRNA-seq datasets are used, and these are mostly confined to cell types from a single tissue. In addition, an easy-to-use and benchmarked tool to conduct the cell type specificity analysis based on GWAS is not available, which hampers replication of published studies.

In this study, we address the current limited use of publicly available scRNA datasets, and propose a workflow to identify cell type specificity of a trait using information from multiple scRNA-seq resources. We systematically curated 43 publicly available scRNA-seq datasets from 32 studies across a variety of tissues/ organs from human and mouse samples. Although direct integration of all the scRNA-seq datasets would be ideal, batch effects induced by differences of protocols seriously complicate

generating a unified scRNA-seq dataset. Several approaches have been introduced aiming to remove such batch effects[22–26], yet these for example, assume the presence of at least one overlapping cell type[25], or require that correlations between cell types are highly similar across datasets[22]. However, there is not always a one-to-one reference between cell types across datasets as the cell type definition largely depends on the clustering methods used in each study[27]. In addition, when integrating multiple datasets, there is a risk of losing unique information that may be highly informative, such as the presence of a cell type in one particular study, but not in others. Because of these limitations, creating a unified reference scRNA-seq set is currently not preferred. Instead, we propose a workflow with cross-dataset conditional analyses to systematically evaluate the association of cell types based on multiple, independent datasets, without the need for direct integration of scRNA-seq expression data and thereby bypassing current hurdles, yet still using information from all datasets. The workflow is implemented in the publicly available web application FUMA[28] (http://fuma.ctglab.nl), which facilitates analyzing cell type specificity for traits (using GWAS results), based on the combined, conditional effects of multiple scRNA-seq datasets. To illustrate the utility of this approach, we applied it to 26 traits covering 6 trait domains for which well-powered GWASs were available. We observed similar cell type association patterns across traits within the same domain, such as endothelial cells in cardiovascular domain, multiple sub-types of neuronal cells in cognitive and psychiatric domains. In addition, we demonstrated that within-dataset and cross-dataset conditional analyses enabled us to identify independent association signals of different cell types.

## Results

**Single cell expression datasets**. Forty-three scRNA-seq datasets were derived from 32 studies[17,18,29–54] for which processed expression data, such as read count, unique molecular identifier (UMI) count, read per kilo base per million (RPKM) or transcripts per kilobase million (TPM) was available (see "Methods" section and Supplementary Data 1 and 2 for details). Figure 1a provides an overview of the curated datasets in terms of the sample tissue types, and the number of available cell types per tissue. Out of the 43 datasets, 11 were based on human samples and 32 were based on mouse samples. There were two relatively large mouse datasets covering a variety of tissues and organs: Tabula Muris[18] and the Mouse Cell Atlas[17], while 29 datasets were specific to brain samples (7 for human and 22 for mouse including embryo and fetal brain samples).

We first evaluated the similarity of scRNA-seq expression profiles between independent datasets by comparing the average expression per gene across cell types, which represents a general gene expression of the dataset. To account for batch effects between datasets, the average expression per dataset was ranked into 100 bins and Spearman's rank correlation was computed for all possible pairs of 43 datasets (see "Methods" section). Correlations across datasets within the same species were significantly higher than between different species (two-sided Mann–Whitney $U$-test $p = 3.1e-5$ and $p = 1.3e-24$ for human and mouse, respectively). This was mainly driven by the high within-species correlation of brain-specific datasets ($p = 7.2e-7$ and $p = 1.3e-36$ for human and mouse). As expected, Fig. 1b shows a clear separation between brain-specific and non-brain-specific datasets in which correlations within brain-specific datasets were significantly higher than correlations between brain-specific and non-brain-specific datasets regardless of the species ($p = 1.0e-82$). In addition, human brain-specific datasets showed significantly higher correlation with mouse brain-specific

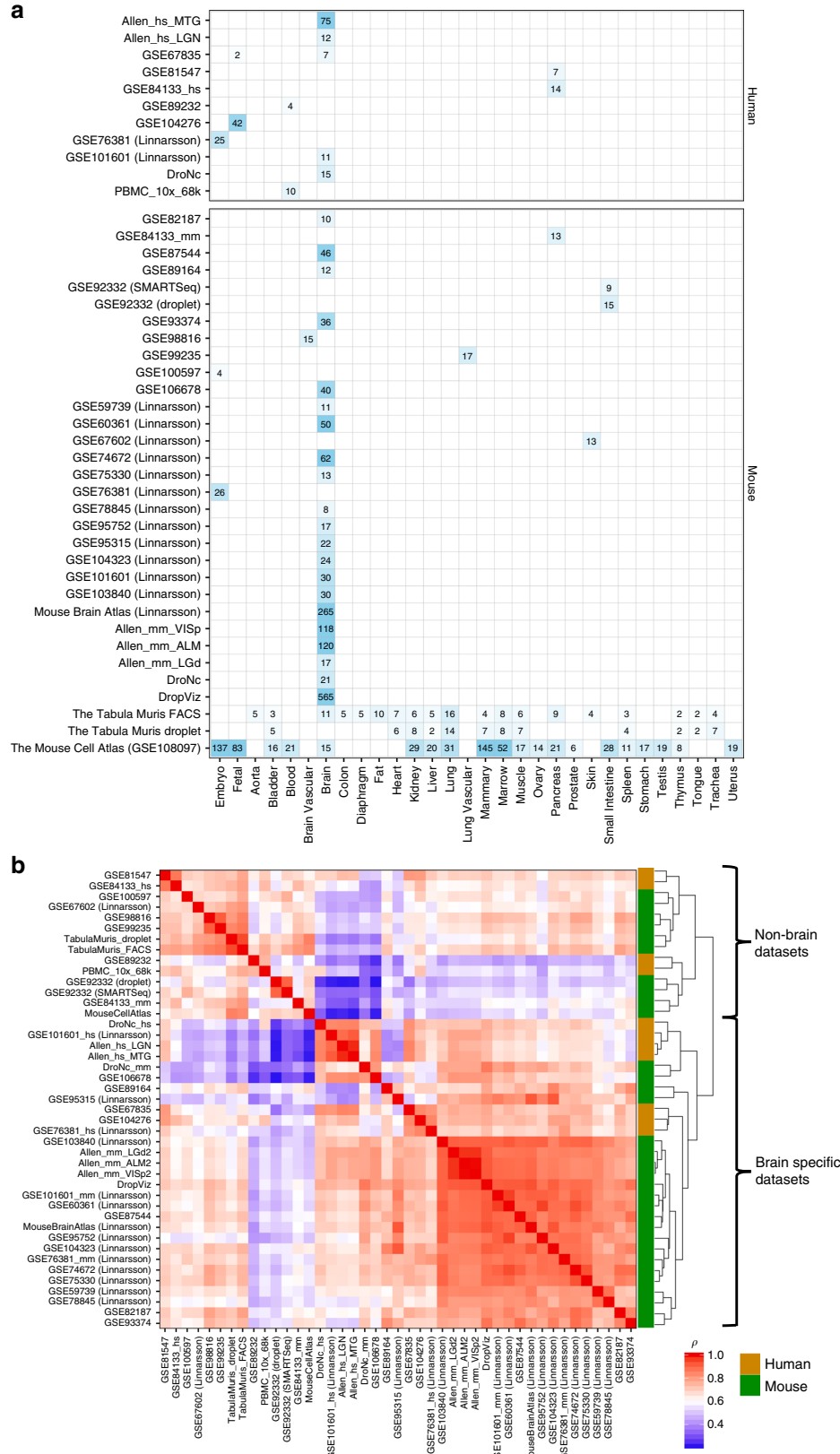

**Fig. 1** Overview of curated scRNA-seq dataset and comparison across datasets. **a** Available tissue types of samples and the number of the cell types defined in each scRNA-seq dataset. The displayed number of cell types is the largest possible number of cell types in the dataset after removing uninformative cell type labels. **b** Pair-wise Spearman's rank correlation of the average expression across cell types between datasets

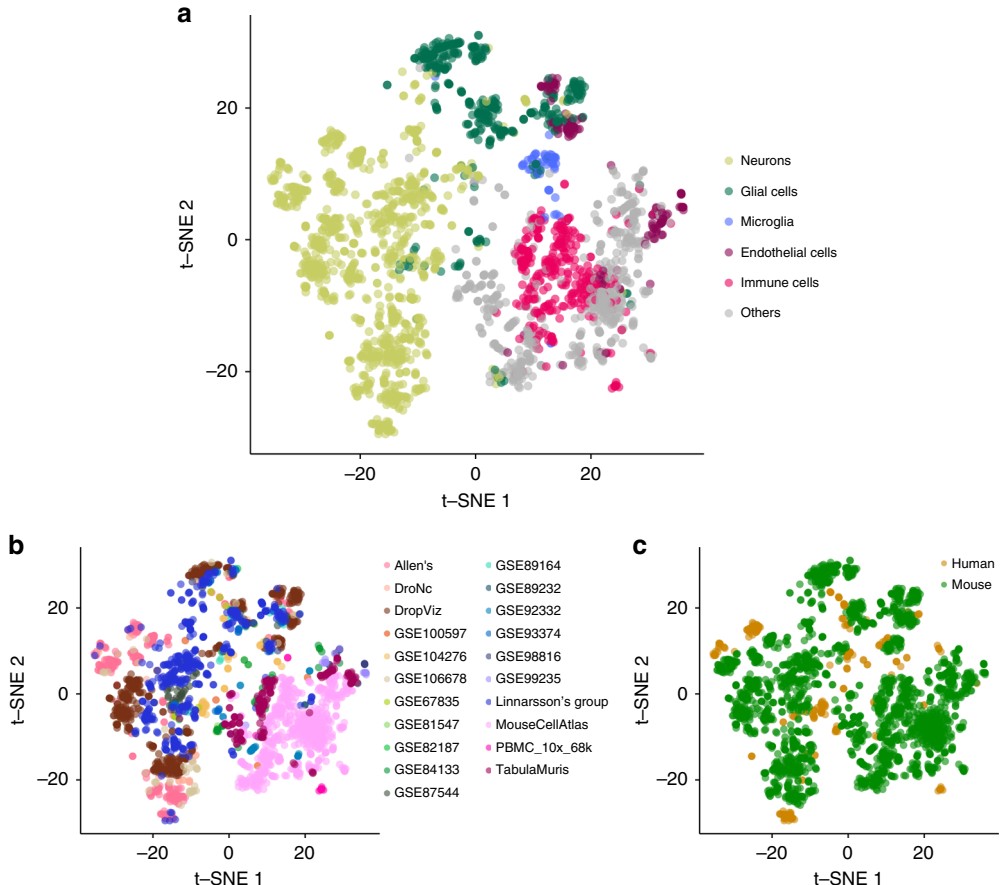

**Fig. 2** 2D projection of cell type similarity based on cell-specific gene expression. Each data point represents a cell type from a dataset. There are 2,679 data points and these are colored by six main categories of cell types (**a**), dataset (**b**), and specie of the samples (**c**). The full results are available in Supplementary Data 3

datasets than with human non-brain datasets ($p = 1.1e-2$), and vice versa for mouse ($p = 8.3e-11$). We also observed significantly higher correlations within non-brain-specific datasets than between brain-specific and non-brain-specific datasets ($p = 2.9e-15$). However, we were not able to evaluate, for non-brain-specific datasets, whether datasets are more strongly correlated within a tissue type than across tissue types due to limited availability.

Next, cell type-specific gene expression patterns were evaluated across the 43 datasets. First, the average expression across cell types within a dataset was regressed out, followed by ranking the residual into 100 bins for each cell type per dataset. For all possible pairs of cell types across 43 datasets, Spearman's rank correlation was computed and projected into a 2D map using t-SNE[55] (see "Methods" section). Each cell type was then manually assigned to one of the six main categories of cell types (i.e. neurons, glial cells, microglia, endothelial cells, immune cells, and others) based on the original cell label (see Supplementary Data 3 for t-SNE coordinates and annotations). For three brain cell types (neurons, glial cells, and microglia), correlations of cell-specific expression profiles were significantly higher within each of the three cell types than across these three brain cell types (two sided Mann–Whitney $U$-test $p < 1.0e-323$ for all three cell types; Fig. 2a). We also found that endothelial cells detected in brain-specific datasets were clustered together with glial cells, and showed a significantly higher correlation with glial cells than with other endothelial cells that were detected in non-brain-specific datasets ($p < 1.0e-323$; Fig. 2a). Cell types from non-brain-specific datasets formed a large cluster distinctively apart from

brain-specific cell types. Within the non-brain-specific cluster, immune cells tend to be clustered together and so did endothelial cells, which showed stronger correlation within each cell type than between other cell types from non-brain-specific datasets ($p < 1.0e-323$ for both immune and endothelial cells). These results show that, across these 43 datasets, cell type-specific expression patterns are highly similar, suggesting the cell types are roughly comparable across datasets, which includes both human and mouse resources. We also observed that the same cell types tended to cluster together across different tissue types (or specific brain regions) of the samples (Supplementary Fig. 1).

We note that, cell types from the same study (or datasets generated from the same lab) tended to cluster together within the same general cell type (Fig. 2b). In addition, cell types from the same species tended to form a cluster within the same category of cell types (Fig. 2c). This clearly indicates the existence of complex batch effects across datasets. In this study, we therefore do not attempt to directly integrate multiple scRNA-seq datasets, but bypass this problem by conducting conditional analyses to allow systematic comparison across datasets.

**Cell type specificity analysis with MAGMA.** To test whether risk variants for a specific trait converge on a specific cell type, we conducted a MAGMA gene-property analysis, where the gene-properties in this case are defined by the gene expression values in a specific cell type. The gene-based $P$-values derived from GWAS results were pre-computed with MAGMA for 26 well-powered ($N > 10,000$) traits from multiple trait domains (see "Methods" section). The gene-property analysis aims to test relationships

between cell type-specific gene expression and disease–gene associations[10,56], which is based on the following regression model:

$$Z = \beta_0 + E_c\beta_E + A\beta_A + B\beta_B + \varepsilon \qquad (1)$$

where $Z$ is a gene-based $Z$-score converted from the gene-based $P$-value (obtained from MAGMA gene analysis), $B$ is a matrix of several technical confounders (such as gene length and correlation between genes based on LD) included by default[56]. $E_c$ is the gene expression value of a testing cell type $c$ and $A$ is the average expression across cell types in a dataset, defined as follows:

$$E_c = \frac{1}{n}\sum_i^n \log_2(e_i + 1) \qquad (2)$$

$$A = \frac{1}{N}\sum_{j \in C}^N E_j \qquad (3)$$

where $n$ is the number of cells in the cell type $c$, $e_i$ is the expression value of a cell in cell type $c$ (e.g. UMI count or CPM), $N$ is the number of cell types in a dataset, and $C = \{$cell type 1, cell type 2, …, cell type $N\}$. In the model, $A$ (average expression across cell types) is added as an additional covariate to identify cell specificity[10] (Supplementary Note 1 and Supplementary Fig. 2). We performed a one-sided test ($\beta_E > 0$) which is essentially testing the positive relationship between cell specificity and genetic association of genes. Note that the $A$ depends on the available cell types and the distribution of them within a dataset, and affects the definition of the cell specificity. This issue is discussed in more detail in the later section.

We compared our MAGMA regression model with the regression model previously applied by Skene et al.[19], where expression values were converted to a binned cell type specificity of gene expression value, which is a proportional expression of a gene in a cell type relative to the sum of all cell types (denoted here as specificity ($S$) score to distinguish from the expression value $E$ used in our model)[19]. We showed that the model of Skene et al. inflates the results of MAGMA gene-property analyses because binned $S$ scores can have strong positive correlations with the average expression across cell types (Supplementary Fig. 3). This suggests that the associations of the binned $S$ score variables are vulnerable to confounding by a general effect of gene expression, making it very difficult to draw reliable conclusions concerning cell type specificity (see "Methods" section, Supplementary Note 2 and Supplementary Figs. 3–5). Therefore, we believe correcting the average expression across cell types independently from the cell-specific expression value is crucial to identify cell specificity.

In addition, there are often genes with expression value zero or very close to zero in all or the majority of cells in a scRNA-seq dataset. It is not always known whether these genes are truly not expressed or are expressed at levels that cannot be accurately measured with current technology. The presence of genes with low-expression values influences the distribution of cell type-specific expression values. The decision how to treat them may unintendingly influence outcomes of cell type-enrichment analyses. The model we propose here showed highly similar results regardless of including or excluding low-expression values, and thus is not influenced by skewed distributions of expression values and does not necessitate excluding low-expression values (Supplementary Fig. 6).

We next compared our approach with two previously proposed methods that aim to identify cell type specificity of a trait using GWAS summary statistics: LD score regression (LDSC)[5] and RolyPoly[6] (see "Methods" section for details). We found that both LDSC and RolyPoly resulted in less significant trait-cell type

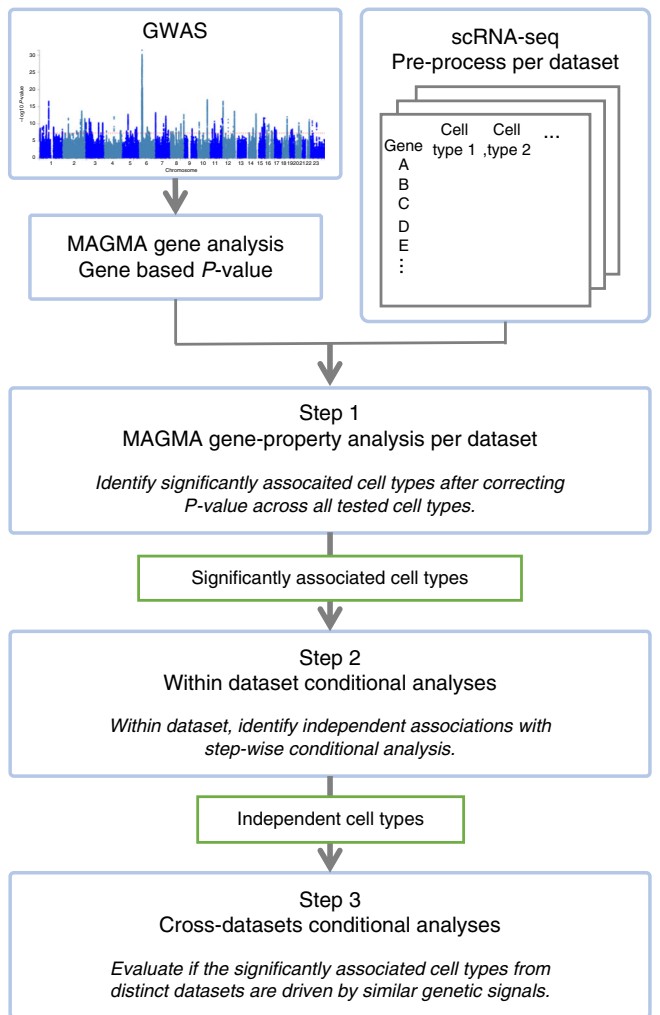

**Fig. 3** Flowchart of cell type specificity analysis using multiple scRNA-seq resources with MAGMA

associations compared to the MAGMA regression model (Supplementary Figs. 7 and 8 and Supplementary Note 3). We therefore conclude that the MAGMA regression model may be preferred. However, we do note that each of these methods tests slightly different hypotheses and it can be informative to compare outputs from multiple tools (Supplementary Note 3).

**A workflow of cell type-specificity analysis.** As mentioned previously, integration of scRNA-seq datasets across multiple studies is challenging due to complex batch and sampling effects. Here we propose a workflow that allows comparison of cell type-specific gene expression profiles from different scRNA-seq datasets that are significantly associated with a trait, using a conditional analysis to avoid the need for direct integration of the gene expression profiles. The workflow consists of three steps as described in Fig. 3.

In the first step, MAGMA cell-specificity analyses are performed for each of the 43 scRNA-seq datasets separately using the regression model described above. Multiple testing correction is applied to the results for all tested cell types across datasets (Supplementary Note 4 and Supplementary Fig. 9).

The second step is a within dataset conditional analysis. It is often the case that there are multiple similar cell types defined in a scRNA-seq dataset, especially when the resolution of cell types is high. The gene expression profiles of those cell types tend to

strongly correlate with each other, and when a cell type is strongly associated with a trait, it is not clear whether that reflects a genuine involvement of that cell type or whether there is confounding due to expression in another cell type correlated with it. The conditional analysis is, therefore, essential to disentangle relationships between trait-associated cell types[10]. In step 2, a systematical step-wise conditional analysis per dataset is performed, by setting thresholds for proportional significance (PS) of the conditional $P$-value of a cell type relative to the marginal $P$-value ($PS_{a,b} = -\log10(p_{a,b})/-\log_{10}(p_a)$, where $p_a$ is the marginal $P$-value for the cell type $a$ and $p_{a,b}$ is the conditional $P$-value of the cell type $a$ conditioning on the cell type $b$). The results will show which cell types to retain or discard from the final step (see "Methods" section and Supplementary Table 1, as well as ref. [10] for details).

The last step is to unravel relationships between significantly associated cell types across datasets. Although the absolute gene expression values in different datasets are not directly comparable, cross-datasets (CD) conditional analysis allows us to test the extent to which the significant gene expression profiles found in different data sets reflect the same or similar association signals. The analysis is performed for all possible CD pairs of significant cell types retained from the second step (see "Methods" section for details). Then the PS of the CD conditional $P$-value of a cell type relative to the CD marginal $P$-value is computed for each cell type of all possible pairs. In this step, the pair-wise conditional analysis provides an overview of independent clusters of signals associated with a trait.

**Application to GWAS summary statistics of 26 traits**. To gain insight into the cell types implicated in a variety of traits, we selected GWAS summary statistics for 26 traits from six disease domains (cardiovascular, immunological, metabolic, cognitive, neurological, and psychiatric traits), which are well powered (sample size > 10,000) and widely studied in terms of both genetics and pathological pathways (Supplementary Table 2).

We first performed MAGMA gene analyses for 20,260 protein-coding genes extracted from Ensembl v92 (GRCh37) with 1 kb windows both sides. Note that we confirmed that gene analysis results are not considerably sensitive to the size of the gene window (Supplementary Fig. 10). We subsequently performed cell type specificity analysis following the three steps described above. For step 1, Bonferroni correction was employed across 43 datasets; in total 2679 unique dataset–cell type combinations were tested, setting the significance threshold at $P_{\text{bon}} = 0.05/2679 = 1.87e-5$. To evaluate the added value of scRNA-seq datasets above the traditional tissue-specific RNA-seq datasets, we computed tissue specificity for these 26 traits using RNA-seq data from GTEx v7 (53 tissue types; see "Methods" section and Supplementary Data 4).

We first evaluated the similarity of cell type association patterns across 26 traits (see "Methods" section). Overall, we observed that traits within the same domain tended to show similar cell-type specificity, except for traits in the metabolic domain, which did not form a single cluster (Fig. 4a). In addition, cell-type association patterns of traits across three specific domains (cognitive, neurological, and psychiatric domains) tended to be very similar. Summarizing the proportion of significantly associated cell types using the six main categories of the cell types, we find that traits in cardiovascular domain tended to show significant association in endothelial and glial cells, immunological domain in immune cells and microglia, and cognitive, neurological and psychiatric domains in neuronal cells (Fig. 4b). Number of significantly associated cell types at each steps and independent signals are summarized in Table 1 for 26

traits and full results are provided in Supplementary Data 5. For each of the clusters of traits, specific cell types were implied as detailed below.

**Cardiovascular domain**. Based on GTEx data, coronary artery disease (CAD), diastolic blood pressure (DBP) and systolic blood pressure (SBP) showed the most significant association with artery tissue, high blood pressure (HBP) with uterus and cervix, and pulse rate (PR) with heart tissue (Supplementary Data 4). From cell specificity analyses, four traits (CAD, HBP, DBP, and SBP) showed strong association with endothelial or other vascular cells from multiple datasets (Table 1 and Supplementary Fig. 11a). Conditional analyses showed that these associations of endothelial cells across multiple datasets were mostly not independent (Fig. 5a and Supplementary Fig. 12a–c), which indicates that the association with endothelial cells is supported by multiple independent datasets. We also observed an association of astrocytes for CAD, supporting the previously reported involvement of astrocytes in CAD[57]. PR showed stronger association with cardiac muscular cells in heart and smooth vascular cells than endothelial cells (Table 1 and Supplementary Fig. 12d).

**Immunological domain**. All of the five traits in this domain (inflammatory bowel disease (IBD), multiple sclerosis (MS), rheumatoid arthritis (RA), systemic lupus erythematosus (SLE), and type 1 diabetes (T1D)) were associated with immune-enriched tissues (e.g. whole blood and spleen) using GTEx data (Supplementary Data 4). These traits all showed, based on scRNA-seq datasets, association with at least one of the subtypes of B cells from multi-tissue datasets (Table 1, Supplementary Figs. 11b and 12e–i). Pathogenesis of B cells autoimmune/immune-mediated disease has been widely reported[58–60]. In addition, we found independent associations of microglial cells with the five traits (Supplementary Fig. 12e–i), providing additional support for the recently implicated role of microglia in autoimmune diseases[61–63].

**Cognitive, neurological, and psychiatric domains**. Traits in cognitive, neurological, and psychiatric domains tended to show strong associations with multiple brain regions based on GTEx data (Supplementary Data 4). Using scRNA-seq datasets, EA, IQ, NEU, and SCZ showed the strongest association with the broadly defined neurons from Tabula Muris FACS (TMF; Supplementary Fig. 11c). These traits also showed association with multiple subtypes of neural cells from a variety of datasets and a large proportion of these associations were either driven by or jointly explained by the broadly defined neurons from TMF (Fig. 5b and Supplementary Fig. 12j–l). However, several subtypes of neurons showed independent signals. For example, SCZ and IQ showed association with an independent cluster of excitatory neurons (Fig. 5b, Supplementary Fig. 12k). SCZ also showed an association with a sub-class of inhibitory neurons in hindbrain samples (HBINH2) independent from excitatory neurons (Fig. 5b). In addition, GABAergic neuron and lateral neuroblast from embryonic brain samples (Linnarsson GSE76381) showed independent cluster in EA, IQ, NEU, and SCZ. Associations of the general neuronal cell types with some of these traits has been previously reported[19–21], however here we show, for the first time, independent associations of specific excitatory, inhibitory, and embryonic neurons.

Major depressive disorder (MDD), insomnia (ISM), and subjective well-being (SWB) showed association with inhibitory neuron from human lateral geniculate, substantia innominate from mouse brain, and neurons in layer 5 from mouse visual cortex, respectively (Supplementary Data 5). For Alzheimer

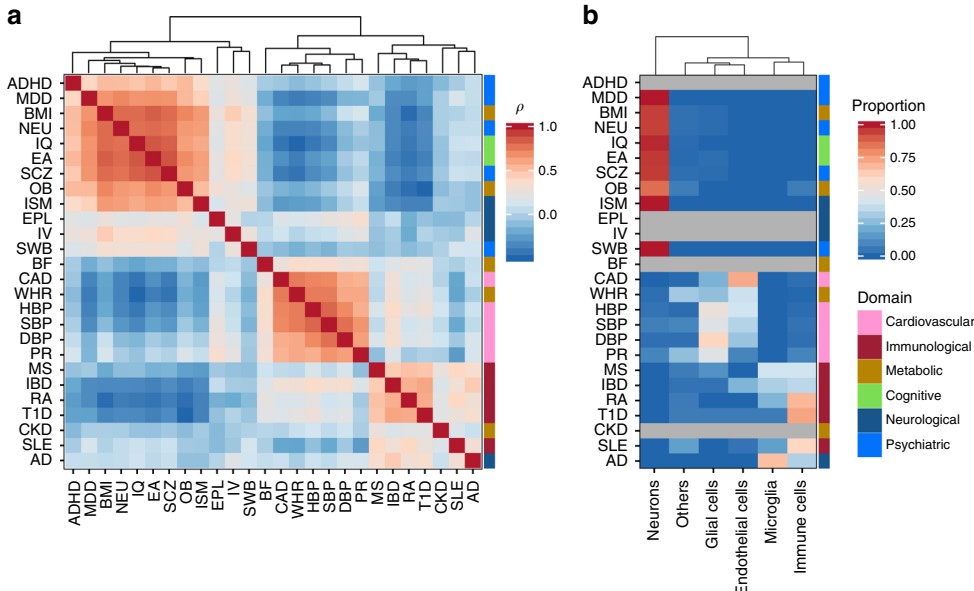

**Fig. 4** Similarity of cell type association patterns across 26 traits. **a** Pair-wise Spearman's rank correlation of cell type association *P*-values from step 1. Traits are clustered based on the pair-wise correlation matrix using the hierarchical clustering. **b** Significantly associated main category of cell types per trait. The heatmap is colored by the proportion of significantly associated cell types ($P < 0.05/2679$) in each category of cell types per trait. Traits with no significant association are colored gray and the traits are in the same order as **a**. The color bar at the right of the heatmap represents the domain of the traits. *P*-values for specific cell types per trait are available in Supplementary Data 5

| Table 1 Summary of cell specificity for 26 traits | | | | | |
|---|---|---|---|---|---|
| **Trait** | **Datasets[a]** | **Step 1[b]** | **Step 2[c]** | **Step 3[d]** | **Summary of independent cell types[e]** |
| *Cardiovascular* | | | | | |
| CAD | 14 | 68 | 16 | 4 | Endothelial cell, astrocytes and muscle cells |
| DBP | 13 | 31 | 16 | 3 | Endothelial cell and brain pericyte |
| HBP | 15 | 45 | 19 | 3 | Endothelial cell, brain pericyte and oligodendrocyte |
| PR | 7 | 11 | 8 | 4 | Cardiac muscle cell, smooth muscle cell, endothelial cell and mural cell |
| SBP | 15 | 53 | 17 | 2 | Endothelial cell and oligodendrocyte precursor cell |
| *Immunological* | | | | | |
| IBD | 22 | 104 | 25 | 4 | Microglia, blood cell, endothelial cell and marrow B-cell |
| MS | 8 | 15 | 10 | 4 | Marrow B-cell, microglia, professional antigen-presenting cell and dendritic cell |
| RA | 17 | 61 | 19 | 5 | Marrow B-cell, blood cell, microglia and professional antigen-presenting cell |
| SLE | 10 | 23 | 12 | 3 | Mammary macrophage, lung B-cell and microglia |
| T1D | 5 | 15 | 5 | 3 | Marrow B-cell, endothelial cell and microglia |
| *Metabolic* | | | | | |
| BF | 0 | 0 | — | — | — |
| BMI | 17 | 147 | 20 | 4 | 4 cluster of neurons |
| CKD | 0 | 0 | — | — | — |
| OB | 4 | 14 | 5 | 5 | 5 cluster of neurons and leukocyte |
| WHR | 21 | 187 | 25 | 6 | Endothelial cell, mesenchymal cell, vascular smooth muscle and outer bulge |
| *Cognitive* | | | | | |
| EA | 16 | 180 | 20 | 5 | 5 clusters of neurons |
| IQ | 16 | 220 | 18 | 4 | 4 clusters of neurons |
| *Neurological* | | | | | |
| AD | 3 | 3 | — | 3 | Monocyte and microglia |
| EPL | 0 | 0 | — | — | — |
| ISM | 1 | 1 | — | — | Substantia innominate |
| IV | 0 | 0 | — | — | — |
| *Psychiatric* | | | | | |
| ADHD | 0 | 0 | — | — | — |
| MDD | 1 | 1 | — | — | Inhibitory neuron |
| NEU | 13 | 87 | 15 | 5 | 5 clusters of neurons and pars tuberalis |
| SCZ | 20 | 187 | 25 | 5 | 5 clusters of neurons |
| SWB | 1 | 1 | — | — | Layer 5 intratelencephaic neuron |

[a]Number of unique datasets of significantly associated cell types from step 1
[b]Number of significant cell types after Bonferroni correction across datasets
[c]Number of cell types retained after step 2 (within dataset conditional analyses)
[d]Number of cell types which are mostly independent across datasets based on step 3 (cross-datasets conditional analyses)
[e]General type of cell types which are mostly independent from step 3, detailed results are available in Supplementary Data 5

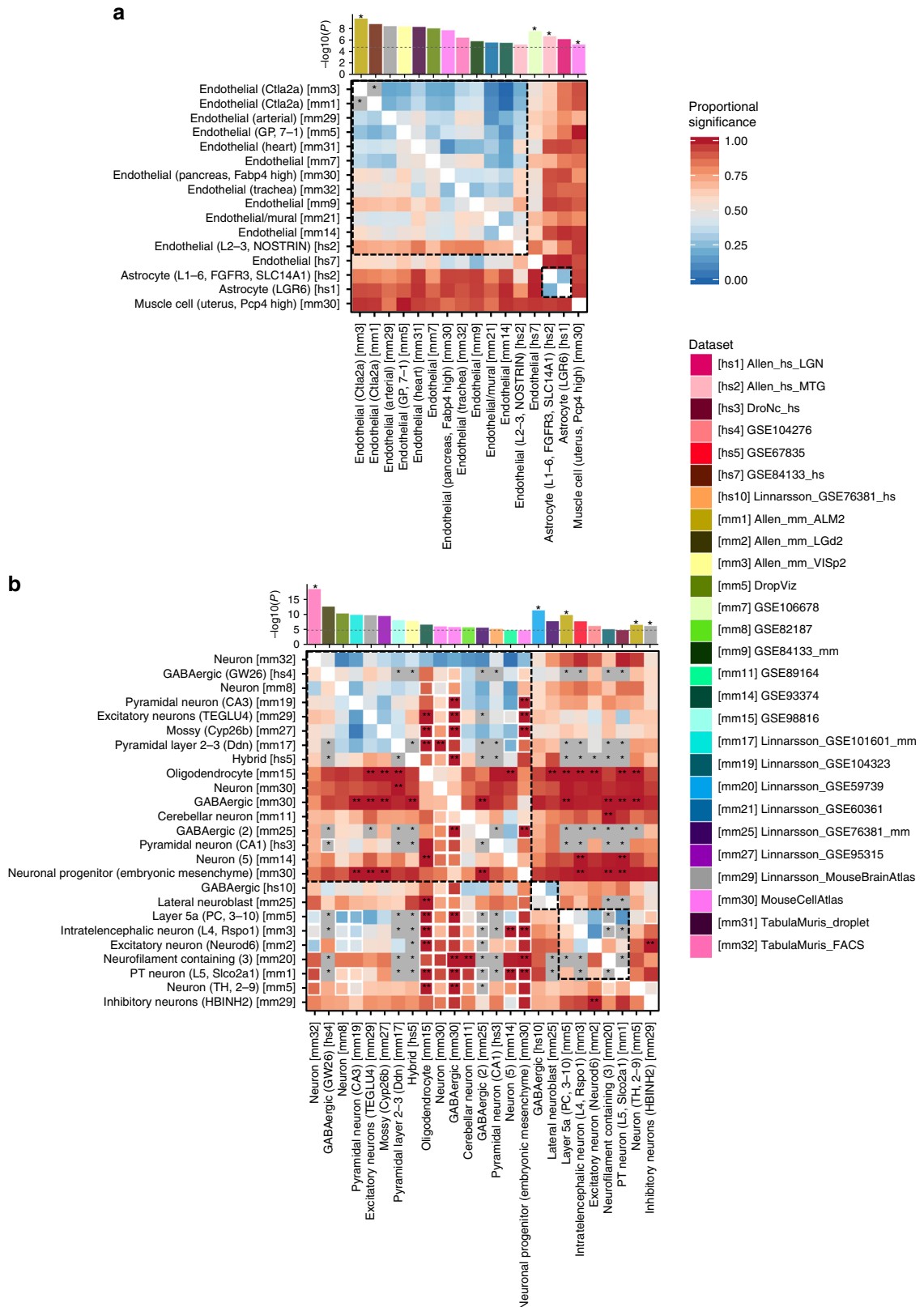

disease (AD), we found a unique cell type association pattern, implicating a role for microglia (Supplementary Data 5). For attention deficit hyperactivity disorder (ADHD), epilepsy (EPL), and intracranial volume (IV), we did not find any significant cell type associations (Supplementary Data 5).

**Metabolic domain.** Of the five traits in the metabolic domain, BMI and obesity (OB) showed association with brain, while WHR showed the strongest association with adipose tissues using GTEx dataset (Supplementary Data 4). Based on cell-specificity analyses, BMI and OB showed very similar cell type association

**Fig. 5** Pair-wise cross-datasets conditional analysis for coronary artery disease (**a**) and schizophrenia (**b**). Heatmap of pair-wise cross-datasets conditional analyses (step 3) for cell types retained from the step 2. Cell types are labeled using their common name with additional information in parentheses (which is needed when referring back to the label from the original study). The index of the dataset is in square brackets. The heatmap is asymmetric; a cell on row $i$ and column $j$ is cross-datasets (CD) proportional significance (PS) of cell type $j$ conditioning on cell type $i$. The CD PS is computed as $-\log_{10}$(CD conditional $P$-value)$/-\log_{10}$(CD marginal $P$-value). The size of the square is smaller (80%) when 50% of the marginal association of a cell type in column $j$ is explained by adding the average expression of the dataset in row $i$ (before conditioning on the expression of cell type $i$). Stars on the heatmap represent pair of cell types that are colinear. Double starts on the heatmap represent CD PS > 1. The bar plot at the top illustrates marginal $P$-value of the cell types on $x$-axis and stars represent independently associated cell types. Cell types are clustered by their independence, and within each cluster cell types are ordered by their marginal $P$-value. For example, there are four independent associations in (**a**) and cell types without a star are not independent from the association of the first independent cell type (with star) on its left. The complete results are available in Supplementary Data 5. The heatmap for other traits are available in Supplementary Fig. 12

patterns with EA, IQ, NEU, and SCZ, which were significantly associated with the broadly defined neurons from TMF (Supplementary Fig. 11d). Involvement of central nervous system in these metabolic traits has been previously suggested[8]. In addition, both BMI and OB showed significant association with inhibitory neurons that are independent from associations with other neuronal cell types (Supplementary Fig. 12m, n). On the other hand, WHR showed distinct cell type specificity compared to other metabolic traits (Fig. 4), with independent signals in endothelial cells, smooth vascular cells, and mesenchymal cells (Supplementary Fig. 12o). Body fat percentage and chronic kidney disease did not show any significant association with any cell type (Supplementary Data 5).

We also provide a step-by-step interpretation of results from the three steps in the analysis workflow, for three specific traits (CAD, IBD, and SCZ) in Supplementary Note 5.

**Effects of general expression across cell types**. In studies integrating scRNA-seq datasets with GWAS summary statistics to identify trait relevant cell types (including this study), the specificity of a cell type is defined by the gene expression of the cell type relative to the general expression of genes within a dataset (e.g. the average expression across cell types)[5,6,19]. Therefore, the cell-type specificity depends on the tested cell types in the dataset, and on the decision which cell types to include in the analysis. For example, the specificity of a cell type in a brain-specific dataset is relative to the overall expression of genes in the brain, while in a multi-tissue dataset, it is relative to a general gene expression across different tissues. To evaluate whether and how the average expression affects trait–cell type associations, we compared associations of neuronal cells and immune cells in the context of differences in cell types included in the analysis (see "Methods" section for details).

We observed decreased significance of the association of neurons from multi-tissue dataset (TMF) when we conditioned on the average across only brain cell types compared to all cell types (Fig. 6a). This explains, for brain-related traits, stronger associations of neurons from TMF compared to associations to neuronal cells from other brain-specific datasets. For a brain-specific dataset, the association of a specific sub-type of an excitatory neuron (TEGLU4 from Mouse Brain Atlas (MBA) dataset) slightly decreased by conditioning on the average expression across only neuronal cell types, while a larger decrease was seen when it was conditioned on the average expression across only excitatory neurons (Fig. 6b). Since 81% of defined cell types in the MBA dataset are neuronal cells, the average across all cell types could be already biased by the neuronal expression, which is why the decrease of significance conditioning on the average of neuronal cells was minimal. Indeed, when a distribution of cell types is balanced (see "Methods" section for details), the significance of the associations increased (Fig. 6c). On the other hand, a large decrease of the significance with

conditioning on the average across all excitatory neurons indicates the association of TEGLU4 is largely confounded by the general expression of excitatory neurons (Fig. 6b).

Similarly, the associations of marrow B-cells from a multi-tissue dataset (TMF) decreased by conditioning on the average across only immune cell types defined in marrow samples (Fig. 6d). This suggests that the association of B-cells with these immunological traits is partially explained by the general expression of immune cells.

To summarize, the strength of the cell type associations is highly sensitive to the definition of the cell type specificity, which depends on the distribution of cell types available in a dataset. This can make the magnitude of the effects highly variable across traits, cell types, and datasets. The conditioning general gene expression changes the definition of the cell type specificity and thereby changes the exact hypothesis that is tested. It is crucial to be aware of what hypothesis is exactly tested given the distribution of cell types available in the datasets.

## Discussion
GWASs have successfully identified genomic loci associated with various human phenotypes, yet so far they provided little insight into the specific cellular mechanisms. Gene expression is a critical intermediate phenotype between genes and functions. The recent advances in sequencing technology provide single-cell resolution of the transcriptome, which allows the collection of information in unprecedented detail. This increase in resolution of cellular taxonomies provides several advantages over the bulk RNA-seq of sampled tissues: first, scRNA-seq enables to single out a specific cell type that is associated with a trait, and second, it increases the power to detect such associations as expression profiles are less heterogenous for single cells than for bulk tissues. By comparing the cell type specific expression patterns across datasets, our results showed that the same cell types tend to cluster together across datasets with different taxonomy or tissue types of the samples.

We note that cell type specificity can be also implicated by other types of data resources, such as ATAC-seq[64] and chromatin markers[65]. We focused on scRNA-seq in this manuscript because of the relatively large availability of the datasets. In addition, the MAGMA gene-property analysis requires annotations per gene while ATAC-seq and chromatin markers are genome-based annotations, although this may not be a problem for approaches, such as LDSC where annotations at SNPs level are used. When data resources such as ATAC-seq and chromatin markers become available more abundantly in single cell resolution, they might also be advantageous for the identification of cell type specificity of a trait.

With increasing availability of scRNA-seq data, combining datasets from multiple independent resources could theoretically reduce dataset specific noise by increasing the number of cells per cell type. However, direct integration of scRNA-seq datasets is

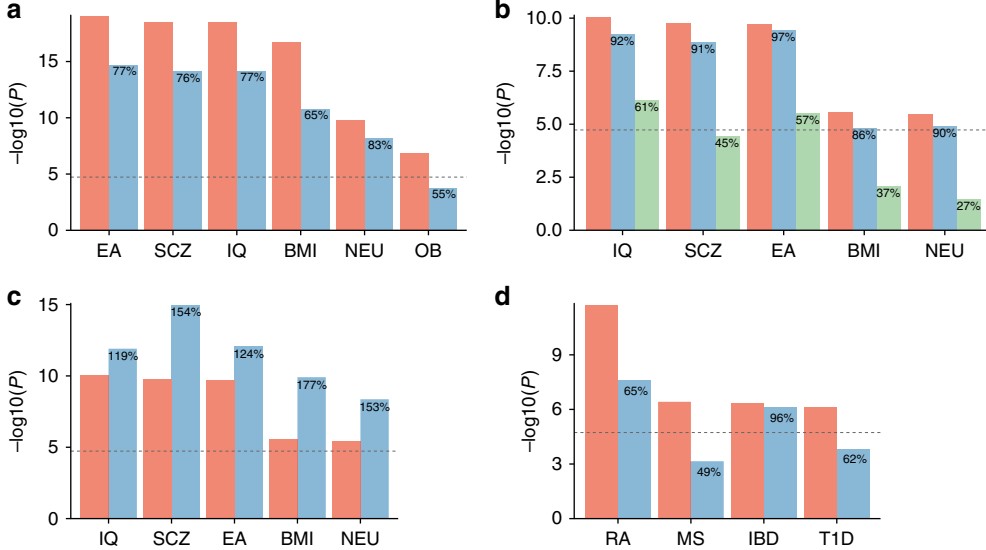

**Fig. 6** Effects of the general expression conditioned in the regression model. **a** Association of *P*-values of neuron from Tabula Muris FACS conditioning on average expression across all available cell types from multiple tissues (pink) or only brain cell types (blue). **b** Association of *P*-values of TEGLU4 (excitatory neuron from cortex) from Mouse Brain Atlas conditioning on average expression across all available cell types (pink), only neuronal cell types (blue), or only excitatory neurons (green). **c** Association of *P*-values of TEGLU4 (subtype of excitatory neurons from cortex) from Mouse Brain Atlas conditioning on average expression across all available cell types (pink) or randomly selected 35 cell types (including TEGLU4) with uniform distribution across seven cell type classes (blue). **d** Association of *P*-values of Marrow B-cell from Tabula Muris FACS conditioning on average expression across all available cell types (pink) or only immune cell types in Marrow samples (blue). The percentages displayed on the blue and green bars represent the proportional significance (in $-\log_{10}$ scale) compared to the pink bars. EA educational attainment, SCZ schizophrenia, IQ intelligence, BMI body mass index, NEU neuroticism, OB obesity, ISM insomnia, RA rheumatoid arthritis, MS multiple sclerosis, T1D type 1 diabetes, IBD inflammatory bowel disease

complicated due to batch and sampling effects. We therefore proposed CD conditional analyses to detect cell types implicated in diseases, while benefiting from the availability of multiple scRNA-seq datasets. By applying this method to 26 traits using 43 scRNA datasets, we demonstrated that multi-step conditional analyses can disentangle relationships between associated cell types which reveal independent signals of specific cell types. For example, EA, IQ and NEU and SCZ showed independent associations of excitatory, inhibitory, and embryonic neurons from the most significant broadly defined neurons.

Our results also showed the advantage of using scRNA-seq over bulk RNA-seq datasets by pinpointing associations of specific cell types with traits. For instance, CAD showed significant associations with artery in coronary and aorta using GTEx dataset, and results of cell type specificity associations suggest that this association is likely to be driven by endothelial cells. In addition, the high resolution of cell type definitions allowed us to pinpoint slightly different association patterns across traits. Indeed, BMI in general showed a very similar cell type specificity pattern with multiple traits from neurological, cognitive, and psychiatric domains, however the strong associations of a subtype of excitatory neurons in the cortex from the MBA dataset (TEGLU) with neurological, cognitive, and psychiatric traits was not retained in BMI. Instead BMI showed stronger association with inhibitory neurons of the hindbrain (HIBNH), which jointly explained the association of TEGLU, while the associations of TEGLU and HIBNH were independent from each other for both EA and SCZ. These differences between BMI and brain-related traits were not distinguishable when examining tissue specificity, demonstrating that scRNA-seq datasets can provide more specific information compared to bulk RNA-seq.

We also highlighted the potential effect of how cell type specificity is defined in the context of which other cell types are included in the analysis. We note that it is important to interpret cell type associations in the context of how cell type specific was

defined, e.g. an association with neurons from multi-tissue or brain-specific dataset, each represents an association with neuron-specific expression given general expression across cell types from multiple tissues or general expression of the brain. We provide pre-processed expression value per cell types (https://github.com/Kyoko-wtnb/FUMA_scRNA_data), to allow users to customize which cell types to include depending on their research questions.

Aligning cell type-specific expression profiles with GWAS results can lead us one step closer to functional follow-up experiments. Current availability of scRNA-seq datasets allow a first glimpse into how genes associated with a trait may exert their influence. However, gene expression in phenotypically identical cells can surprisingly vary during the lifespan, and identifying cellular subtypes not only at spatial but also at temporal resolution will also be important in understanding how cellular functions may affect the risk of disease throughout development. In the future, large-scale efforts, such as the Human Cell Atlas, hold a great promise for providing a comprehensive overview of cell-specific gene expression in most of human tissues with much larger sample sizes[16].

## Methods

**URLs**. GEO: https://www.ncbi.nlm.nih.gov/geo/, Linnarsson's group: http://linnarssonlab.org/, Mouse Brain Atlas: http://mousebrain.org/, Tabula Muris: https://hemberg-lab.github.io/scRNA.seq.course/tabula-muris.html, Allen Brain Atlas: http://celltypes.brain-map.org/download, The Mouse Cell Atlas: http://bis.zju.edu.cn/MCA/, DropViz: http://dropviz.org/, Broad single cell portal: https://portals.broadinstitute.org/single_cell, GTEx: https://www.gtexportal.org/, MAGMA: https://ctg.cncr.nl/software/magma, CARDIoGRAMplusC4D: http://www.cardiogramplusc4d.org/, Immunobase: https://www.immunobase.org/downloads/protected_data/GWAS_Data/, GIANT consortium: https://portals.broadinstitute.org/collaboration/giant/index.php/GIANT_consortium_data_files, IGAP: http://web.pasteur-lille.fr/en/recherche/u744/igap/igap_download.php, ENIGMA: http://enigma.ini.usc.edu/, SSGAC: https://www.thessgac.org/, PGC: https://www.med.unc.edu/pgc, FUMA: http://fuma.ctglab.nl/

**Curation and pre-processing of scRNA-seq datasets**. Single-cell RNA-seq (scRNA-seq) datasets were curated from NCBI GEO and other resources (see URLs). We first selected datasets with a total number of cells > 250 and when samples are *Homo sapiens* or *Mus musculus* . We further selected datasets in which both the processed expression data and pre-assigned cell label was available. Processed expression data could be read count, UMI count, RPKM, or TPM. We did not perform any pre-analysis on FASTQ files.

Each dataset was processed separately according to the following steps: (1) When the obtained value was the read count, the count was converted into the count per million (CPM) to allow correction for the total number of reads per cell. Other values, UMI count, RPKM, and TPM were used as is. (2) Quality control (QC) of cells was performed as described in the original study unless the obtained dataset was already QC-ed. (3) Cells with uninformative cell type labels (e.g. 'unclassified' or missing cell label) that are defined as outliers in the original study were excluded with an exception to 'unknown' cell clusters in Tabula Muris datasets which have a potential of being novel cell types. (4) The expression value (UMI count, CPM, RPKM, or transcript per million (TPM)) was log2 transformed with pseudo-count 1 and the per gene per cell type average was computed. (5) Genes provided in the processed datasets were mapped to human Ensembl gene ID (v92). Mouse genes were mapped to human genes using BioMart. Genes which were not mapped to a human Ensembl gene ID were excluded. For all datasets, cell labels were used as provided in the original dataset. Further details and exceptions for each dataset are described in Supplementary Data 2. Then, for each dataset, average expression per cell type was computed with additional column of average across cell types. When there are multiple levels of cell labels, average expression was computed for each level which created multiple files. Similarly, when there are multiple tissue types or developmental stages are available, one average expression file with all cell types from all tissues/developmental stages and multiple files for each tissue/developmental stage separately were created for a single scRNA-seq dataset.

**Mapping genes to human Ensembl gene ID**. We used Ensembl v92 as the primary gene ID in this study. When gene symbols are annotated to expression value in human scRNA-seq dataset, gene symbols were mapped to Ensembl gene ID by matching 'external_gene_name' from BioMart. Genes with duplicated Ensembl gene ID were filtered out. For mouse gene, first mapped mouse Ensembl gene ID to human Ensembl gene ID using BioMart and only genes with one-to-one assignment between human and mouse were extracted. Then mouse gene symbols and aliases are assigned to Ensembl gene ID using NCBI gene information (Mus_musculus.gene_info.gz). Gene symbols in mouse scRNA-seq datasets were then mapped to human Ensembl gene ID by matching either symbol or any of aliases and genes with duplicated Ensembl ID were excluded.

**Datasets and cell types used in analyses**. For each of 43 datasets, we used cell type labels at the highest possible resolution when multiple levels are available. For example, when level 1 and level 2 cell type annotations are available (level 2 defines greater number of cell types), level 2 label was used. For GSE93374, level 2 was used instead of level 3, since level 3 cell type annotation is only assigned to neuronal cells. Similarly, for GSE74672, level 1 was used instead of level 2 which was only for neuronal cells. For GSE60361, level 1 was used instead of level 2, since there are 189 cells without informative label at level 2. This resulted in 2,679 unique cell type–dataset pairs.

**Comparison of scRNA-seq expression profiles across datasets**. To compare general gene expression profiles across datasets, first, the average expression across available cell types per dataset was computed. Then, for any pair of datasets, the correlation coefficient was computed by taking genes present in both datasets and ranking genes into 100 bins to account for batch effects. Genes with an expression value of zero were kept as zero, therefore, there are 101 values in the end (from 0 to 100).

To compare cell type specificity across datasets, expression values of each cell type were regressed on the average expression across cell types per dataset and residuals were used as cell type-specific expression. For each pair of cell types both within and between datasets, correlation coefficient was computed by taking genes present in both datasets and ranking genes into 100 bins as described above. Similarity of cell types was visualized by projecting on to 2D map by performing t-SNE[55] on average expression per cell type per dataset with perplexity 30. t-SNE was performed 100 times for each dataset and optimal projection was obtained by minimizing Kullback–Leibler divergence.

**Comparison with LDSC and RolyPoly**. We used the Tabula Muris FACS dataset including 119 cell types to compare cell type specificity based on different methods for three traits; CAD, IBD, and SCZ as examples. To perform cell type specificity analyses with LDSC, genes are required to have binary state; i.e. they are either specific to a cell type or not. As previously applied by Skene et al.[19], the top 10% of genes with the highest *S* score were defined as cell type-specific genes for each of 119 cell types. We also defined another set of cell type-specific genes by taking the top 10% of genes with the highest residuals after regressing out the average expression across cell types from gene expression per cell type. Then SNPs

within 1 kb windows (both sides) were mapped to cell type-specific genes. We corrected for 53 baseline annotations[66]. To account for presence of genes in TMF datasets, one additional annotation by annotating 1 for SNPs located within 1 kb window of any of genes available in TMF dataset was included for each of 119 cell type.

The input of RolyPoly is a matrix with genes in rows and cell types in columns and it is recommended to scale the expression value across the dataset[6]. We therefore used the average log2-transformed expression per gene per cell type and further scaled across the dataset. Since the software does not allow negative values as an input, the scaled expression values were shifted by the minimum value so that all genes in all cell types have positive value. The RolyPoly implements bootstrap to compute specificity of the cell type which is computationally very expensive and time consuming. Therefore, we limited the analyses to chromosomes between 10 and 22. To compare with the MAGMA regression model, we re-run MAGMA gene-property analyses only using genes on chromosomes 10–22.

**GTEx expression data**. GTEx v7 TPM was obtained from GTEx portal (URLs). Genes with average TPM per tissue > 1 in at least one tissue were included. TPM was log2-transformed with pseudocount 1 after winsorizing at 50, and average expression was computed for each of 53 tissue types.

**MAGMA gene and gene-property analyses**. MAGMA gene analyses were performed for all GWAS summary statistics with matched reference panel, i.e. 1000 Genomes[67] (1000G) or UK Biobank release 2[68] (UKB). Reference panel used for gene analyses are labeled in Supplementary Table 2 for each trait. For 1000G, 504 European subjects were used. For UKB, randomly selected 10,000 unrelated European subjects were used. Although some meta-analyzed GWAS summary statistics contain multiple populations, the majority of subjects were of European ancestry, and we used European population as a reference panel. To maximize the matching of SNPs between GWAS summary statistics and reference panel, unique SNP ID (consists of chromosome:position:allele1_allele2 where alleles are alphabetically ordered) was used instead of rsID. SNPs were assigned to one of the 20,260 coding genes from Ensembl v92 with 1 kb window for both sides. Gene analysis was performed with SNP-wise mean model[56].

MAGMA gene-property analyses (v1.07) were performed using the output of gene analysis and gene expression datasets processed as described above. Average expressions per cell/tissue type were provided as gene covariates to magma by the following flag, '–gene-cov < file name > –model condition-hide = Average direction = greater' which perform one-side test conditioning on average expression across cell/tissue types. For Supplementary Figs. 3–6 where binned *S* score was used as expression value, 'condition-hide = Average' option was excluded. Similarly, for Supplementary Figs. 2–4 where we did not condition on average expression across cell/tissue types, 'condition-hide = Average' option was excluded.

**Within dataset conditional analysis**. In the second step of the workflow, step-wise conditional analyses were performed, per dataset, from the cell type with the most significant marginal *P*-value. In summary, forward selection (retain the cell type with the lowest marginal *P*-value) was performed for a pair of cell types which were jointly explained ($PS_{a,b} < 0.2$ and $PS_{b,a} < 0.2$) or one association was mainly driving the other's ($PS_{a,b} \geq 0.5$ and $p_{b,a} \geq 0.05$, or $PS_{a,b} > 0.8$ and $PS_{b,a} < 0.5$). In the case of partially joint associations ($PS_{a,b} \geq 0.5$ and $PS_{b,a} \geq 0.5$) or independent ($PS_{a,b} \geq 0.8$ and $PS_{b,a} \geq 0.8$), both cell types were retained. Note that when associations of two cell types are jointly explained, only one cell type with the lowest marginal *P*-value is retained for the third step. However, this does not mean the discarded cell type is less important than the retained cell type, but the result suggests that the associations of these two cell types cannot be distinguished. Although conditional *P*-values are often proportional to marginal *P*-values, it is possible that cell type with higher marginal *P*-value results in less conditional *P*-value for a pair of cell types (i.e. $p_{b,a} < p_{a,b}$). Therefore, when $PS_{a,b} < 0.2$ and $PS_{b,a} \geq 0.2$, the order of cell types was flipped for forward selection.

Although only retained cell types were used for the third step, the results of within-dataset conditional analyses for any pair of cell types were further broken down into eight categories, as described in Supplementary Table 1. This is to provide better understanding of the relationship of two significantly associated cell types. For example, in both scenario 4 and 5, cell type B is dropped and cell type A is considered as the main driver of the association. However, in scenario 4, association of cell type B cannot be completely explained by cell type A as conditional *P*-value of cell type B is still < 0.05. Therefore, there might still be a unique signal to cell type B, however, as large amount of significance is dropped, the cell type B is not retained for the further step.

**CD conditional analysis**. For each pair of cell types from different datasets, the following three regression models were tested to incorporate the effect of the average expression from the other dataset:

$$Z = \beta_0 + E_{c1}\beta_{E_{c1}} + A_1\beta_{A_1} + A_2\beta_{A_2} + B\beta_B + \varepsilon \qquad (4)$$

$$Z = \beta_0 + E_{c2}\beta_{E_{c2}} + A_1\beta_{A_1} + A_2\beta_{A_2} + B\beta_B + \varepsilon \qquad (5)$$

$$Z = \beta_0 + E_{c1}\beta_{E_{c1}} + E_{c2}\beta_{E_{c2}} + A_1\beta_{A_1} + A_2\beta_{A_2} + B\beta_B + \varepsilon \qquad (6)$$

where $E_{cx}$ is an average log-transformed expression of cell type $c$ from dataset $x$, and $A_x$ is an average expression across cell types in dataset $x$. In this step, we define $P$-value of testing alternative hypothesis $\beta_{Ecx} > 0$ from 1st and 2nd models as CD marginal $P$-value, and $\beta_{Ecx} > 0$ from 3rd model as CD conditional $P$-value for a cell type $c$ from a dataset $x$. Note that, when associations of two cell types from different datasets with a trait are largely disappeared by conditioning on each other, it suggests that associations of those cell types were driven by similar genetic signals but this does not measure the similarity of two cell types (i.e. it cannot be concluded that the cell types from the different datasets are the same). To summarize the results from step 3, we defined independently associated cell types based on forward-selection (by ordering the cell types with marginal $P$-value), where we considered cell types with PS > 0.5 on each other are independent.

**Effects of the average gene expression across cell types**. To compare average expression across all cell types and only brain cell types, we used TMF (multi-tissue) dataset. For brain-specific expression, 9 cell types with the label of tissue 'Brain Non_Myeloid' or 'Brain Myeloyd' in the original dataset were extracted. We used six traits which showed significant marginal association with the neurons for the comparison.

To compare average expression across all brain cell types and only neuronal cell types, we used MBA dataset. For neuron-specific expression, 214 neuronal cell types with 'Neuron' in the highest class in the original dataset were extracted. For excitatory neuron-specific expression, 60 excitatory neurons which contained a word 'excitatory' in the description of the cell type in the original dataset was extracted. To create a dataset with cell types evenly distributed across the highest class of the brain cell types, five-cell types were randomly selected for each of seven classes defined in the MBA dataset (i.e. astrocytes, ependymal, immune, neurons, oligodendrocytes, peripheral glia, and vascular), resulted in 35-cell types including TEGLU4. We used five traits that showed significant marginal $P$-value with TEGLU4 for the comparison.

To compare average expression across all cell types and only immune cell types, we used TMF dataset. For immune cell-specific expression, 21-cell types from marrow samples were extracted. We used four immunological traits which showed significant marginal $P$-value with B cell.

**Binning of expression value for model comparison**. To compare our regression model with the model used in Skene et al. we created binned $S$ score as described in their original study[19], $S$ score was binned into 40 bins by equally binning genes based on average expression (each gene gets a value between 1 and 40) and genes with $S$ score 0 were kept as it is. Therefore, by including 0 there are 41 bins in total. For Supplementary Fig. 3, we binned log-transformed average expression per cell type into 40 bins in the same way as $S$ score. For Supplementary Fig. 4, we changed the number of bins to 20, 100, 1000, and the number of genes with $S$ score (or log-transformed expression value) > 0.

**Reporting summary**. Further information on research design is available in the Nature Research Reporting Summary linked to this article.

## Data availability
Pre-processed scRNA-seq datasets are available at: https://github.com/Kyoko-wtnb/FUMA_scRNA_data.

## Code availability
Scripts for pre-processing of scRNA-seq datasets are available at: https://github.com/Kyoko-wtnb/FUMA_scRNA_data. Implementation of MAGMA gene-property analysis on FUMA web application is available at: https://github.com/Kyoko-wtnb/FUMA-webapp.

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

## Acknowledgements

This work was funded by The Netherlands Organization for Scientific Research (NWO VICI 453-14-005). We thank the GIANT consortium, CARdioGRAMplusc4D, SSGAC, IGAP, ENIGMA, and PGC for providing GWAS summary statistics and GTEx Portal for RNA-seq data. We also thank Tabula Muris consortium and other individual groups for making scRNA-seq datasets publicly available.

## Author contributions

D.P. conceived the study; K.W. performed the analyses; M.U.M. supported the analyses; C.d.L. and M.P.v.d.H. provided statistical advice and discussion; K.W., M.U.M., and D.P. wrote the paper.

## Additional information

**Competing interests:** The authors declare no competing interests.

