## [Peer Review File · Nature Communications]

Editorial Note: This manuscript has been previously reviewed at another journal that is not operating a transparent peer review scheme. This document only contains reviewer comments and rebuttal letters for versions considered at Nature Communications .

Reviewers' comments:

Reviewer #1 (Remarks to the Author):

The manuscript is greatly improved. I particularly appreciate the author's attention to the issue of robustness of results to the set of cell types included in an analysis, and the new framework of conditional analyses. I have only two remaining comments.

1. The text and figures are still lacking clarity. In particular, I would hope to have a clear visual answer, in the figures, to the question "which cell types show signal for which phenotypes after steps 1, 2, and 3 of the pipeline?" This information is encoded in Figures 4b, 5, and 6, but is nearly impossible to extract.

- For Figure 4b, what is in the clusters of cell types? Even a sentence of interpretation in the legend would be helpful, since the colored dendrogram makes it look like they are all a mix of many things.

- Figure 5 is referenced after sentences such as "[CAD, HBP, DBP, and SBP] showed strong association with endothelial or other vascular cells from multiple datasets." But in order to extract this message from Figure 5 requires reading small axis labels and then going into the supplement to figure out what e.g. VECA or SMC_Acta2 is. Coloring is by dataset and is not very helpful: we are more interested in cell type than dataset, and in any case the datasets are labeled cryptically in the legend (e.g. Linnarsson_GSE74672) and the colors are too similar to decipher. Using color or order of cell types along the x axis to make it easier to verify the claims in the main text would be hugely helpful here.

Re-labeling the cell types with interpretable names is also key.

- For Figure 6, is there a way to summarize which cell types you are considering independent signals and which are being supported across multiple datasets?

- The section on the Skene et al method is clearer now, but I would recommend being clearer about what is inflation and what is type 1 error. For example, if your null hypothesis includes non-cell-type-specific general enrichment, then there is type 1 error. How about leaving out discussion of type 1 error and instead saying that there are significant results due to non-cell-type-specific enrichment, but there are no significant results in the absence of enrichment altogether.

2. The comparisons to S-LDSC and RolyPoly strengthen the manuscript. However, the comparison to S-LDSC does not use that method as it was applied to bulk gene expression data (Finucane et al 2018). In particular, Finucane et al. 2018 used regression coefficients instead of total enrichment to score cell types, and showed that type 1 error was controlled in simulation. The authors should compare to S-LDSC as it is used for gene expression data.

Reviewer #2 (Remarks to the Author):

The authors mostly addressed my previous specific comments. One specific comment of mine which I felt the authors missed the point on was the relationship of their approach to approaches using single-cell ATAC-seq data to implicate specific cell types associated with GWAS. I was previously suggesting the authors should consider having some discussion on the relative merits of implicating cell-types directly from single cell ATAC-seq vs. using single cell gene expression data. Cusanovich et al, Cell 2018 showed that specific cell types could be implicated directly without needing to associate ATAC-seq signal with genes. I was not proposing, as the authors seem to suggest in their response of assigning ATAC-seq to specific genes to conduct enrichment analyses. For the problem that the

authors motivate of identifying the relevant cell type for GWAS, that step does not seem necessary if one uses analysis approaches such as LDSC in conjunction with ATAC-seq data as shown in Cusanovich et al, Cell 2018.

I was also asked to comment on the comments of Reviewer 3. I am not sure I fully understand the reviewer's previous point on the Skene's et al model. The authors have added a comparison with RolyPoly as well as stLDSC, which also addressed a comment I previously raised. My overall impression from the comparisons with these previous models is the authors model is doing something different and reasonable, but a ground truth is lacking to be able to make definitive statements of the relative sensitivity and specificity of different approaches.

In terms of the general issue of the significance of the biological results raised both in my review and of reviewer 3, I agree many of the observations were well established, though the authors do also point out some more novel/specialized results. I see the issue more being to what extent one can be confident about more novel and specialized results since they seem to be more dependent on the specifics on how the analysis is conducted. This issue is seen with the removal of the novel result of the relationship between pancreatic cells and brain-related traits, highlighted by reviewer 1, between the original submission and this revised submission. Some of the results are also dependent on the specifics of which cell types are being conditioning on, something which the authors are more transparent about in the revised version.

More generally, the issue was raised both in my review and of reviewer 3 of the overall significance of this work. My overall impression of this manuscript is that it is a collection of incremental advances/observations (e.g. analyzing a large number of single-cell RNA-seq data, incorporating in a web-tool, the technical differences between the Skene et al and their model, the added workflow for conditional analyses), but I can also see the argument that collectively this work then becomes significant and will be useful to people.

Below we provide a point-by-point response indicating the changes made to the manuscript. For clarity, our response is written in blue.

We would like to briefly note that, there was a minor bug in some of our scripts which caused a mismatch of MAGMA versions between some analyses (the latest version v1.07 was used in this manuscript while v1.06 was used in the initial submission). We therefore updated some of the association statistics (and minor changes in the main text), however the nature of the results remained the same and the conclusions of the manuscript did not change. All changes due to this update are highlighted in blue in the main text and Supplementary Note.

Reviewer #1 (Remarks to the Author):

The manuscript is greatly improved. I particularly appreciate the author's attention to the issue of robustness of results to the set of cell types included in an analysis, and the new framework of conditional analyses. I have only two remaining comments.

1. The text and figures are still lacking clarity. In particular, I would hope to have a clear visual answer, in the figures, to the question "which cell types show signal for which phenotypes after steps 1, 2, and 3 of the pipeline?" This information is encoded in Figures 4b, 5, and 6, but is nearly impossible to extract.

We apologize for the lack of clarity and agree this could be improved. As suggested, we have now changed Fig. 4b, 5 and 6 to increase clarity and facilitate visual inspection of a summary of results. Briefly, we re-labelled the cell types in the plots with more common names. We do provide the original labels in the Supplementary Tables which allows to refer back to the original study. We also re-ordered the cells in Fig. 6 (heatmap of results from step 3, which is Fig. 5 in the revised version) to make it much easier to capture which cluster of cell types showed independent associations. In addition, we added Table 1 in the main text which includes a summary of results from each step for each trait with common names of cells that showed independent associations. Details for each of these points are described below.

- For Figure 4b, what is in the clusters of cell types? Even a sentence of interpretation in the legend would be helpful, since the colored dendrogram makes it look like they are all a mix of many things.

We agree there may have been too much information in this plot. We now updated **Fig. 4b** and, instead of displaying P-values of 158 cell types which showed significant association with at least one of the 26 traits, we now display this for the 6 main categories of cell types, and include information (the color bar) on the proportion of significantly associated cell types within each of the 6 categories, for each trait. We also modified the main text and legend accordingly.

- Changes in the main text (from line 252)
Summarizing the proportion of significantly associated cell types using the six main categories of the cell type, we find that traits in cardiovascular domains tend to show significant association in endothelial and glial cells, immunological domains in immune cells and microglia, and cognitive, neurological and psychiatric domains in neuronal cells (**Fig. 4b**).
- Changes in **Fig. 4b**

- **Fig. 4 legends**
Figure 4. Similarity of cell type association patterns across 26 traits. (a) Pair-wise Spearman's rank correlation of cell type association P-values from step 1. Traits are clustered based on the pair-wise correlation matrix using the hierarchical clustering. (b) Significantly associated main category of cell types per trait. The heatmap is colored by the proportion of significantly associated cell types ($p < 0.05/2679$) in each category of cell types per trait. Traits with no significant association are colored grey and the traits are in the same order as (a). The color bar at the right of the heatmap represents the domain of the traits. P-values for specific cell types per trait are available in **Tables S7-84**.

- Figure 5 is referenced after sentences such as "[CAD, HBP, DBP, and SBP] showed strong association with endothelial or other vascular cells from multiple datasets." But in order to extract this message from Figure 5 requires reading small axis labels and then going into the supplement to figure out what e.g. VECA or SMC_Acta2 is. Coloring is by dataset and is not very helpful: we are more interested in cell type than dataset, and in any case the datasets are labeled cryptically in the legend (e.g. Linnarsson_GSE74672) and the colors are too similar to decipher. Using color or order of cell types along the x axis to make it easier to verify the claims in the main text would be hugely helpful here. Re-labeling the cell types with interpretable names is also key.

We thank the reviewer for this useful suggestion. Due to the limited space and the number of cells in the plot, we now moved the previous **Fig. 5** to **Supplementary Fig. 11** so that the figure can be large enough to read cell labels. Although we decided to keep the color based on the dataset (since this information is important for step 2 and 3), we added a dataset index in the cell type label, in case the color is not distinguishable. In this way readers can still identify the original datasets. Regarding the cell labels, we now replaced these with the more common name of the cell types, and kept additional information in parentheses, which are necessary to refer back to the original cell labels from the original datasets. Since we moved **Fig. 5** to the Supplementary Materials, we now added **Table 1** to the main text to summarize the results of cell type associations across 26 traits.

- Changes in **Supplementary Fig. 11**
- Additional **Table 1** in the main text

- For Figure 6, is there a way to summarize which cell types you are considering independent signals and which are being supported across multiple datasets?

We now labelled independently associated cell types with stars in the figures and clustered cell types by their independence, ordered by the marginal P-value within the cluster. This should make it is easier to capture what is independent and which cell types belong to which cluster. This is now also explained clearer in the legend.

- Changes in the figures
Fig. 5
Supplementary Fig. 12

- **Fig. 5 legend**
Figure 5. Pair-wise cross-datasets conditional analysis for coronary artery disease (a) and schizophrenia (b). Heatmap of pair-wise cross-datasets conditional analyses (step 3) for cell types retained from the step 2. Cell types are labeled using their common name with additional information in parentheses (which is needed when referring back to the label from the original study). The index of the dataset is in square brackets. The heatmap is asymmetric; a cell on row i and column j is cross-datasets (CD) proportional significance (PS) of cell type j conditioning on cell type i . The CD PS is computed as $-\log_{10}(\text{CD conditional P-value}) / -\log_{10}(\text{CD marginal P-value})$. The size of the square is smaller (80%) when 50% of the marginal association of a cell type in column j is explained by adding the average expression of the dataset in row i (before conditioning on the expression of cell type i). Stars on the heatmap represent pair of cell types that are colinear. Double stars on the heatmap represent $\text{CD PS} > 1$. The bar plot at the top illustrates marginal P-value of the cell types on x-axis and stars represent independently associated cell types. Cell types are clustered by their independence, and within each cluster cell types are ordered by their marginal P-value. For example, there are 4 independent associations in (a) and cell types without a star are not independent from the association of the first independent cell type (with star) on its left. The complete results are available in **Supplementary Table 10** and **63**. The heatmap for other traits are available in **Supplementary Fig. 12**.

- The section on the Skene et al method is clearer now, but I would recommend being clearer about what is inflation and what is type 1 error. For example, if your null hypothesis includes non-cell-type-specific general enrichment, then there is type 1 error. How about leaving out discussion of type 1 error and instead saying that there are significant results due to non-cell-type-specific enrichment, but there are no significant results in the absence of enrichment altogether.

Following the suggestion of this reviewer, we have now left out the part of type 1 error from **Supplementary Note 2**.

- Changes in **Supplementary Note 2**

2. The comparisons to S-LDSC and RolyPoly strengthen the manuscript. However, the comparison to S-LDSC does not use that method as it was applied to bulk gene expression data (Finucane et al 2018). In particular, Finucane et al. 2018 used regression coefficients instead of total enrichment to score cell types, and showed that type 1 error was controlled in simulation. The authors should compare to S-LDSC as it is used for gene expression data.

Thanks for pointing this out. We have now re-run LDSC with the correct flag (--h2-cts) as suggested, which tests if the regression coefficient is greater than zero. We found that, although some of top significant cell types were consistent between LDSC and MAGMA, MAGMA tends to identify a larger number of significant cell types. We have modified the main text and supplementary note accordingly.

- Changes in the main text (from line 191)
 We found that both LDSC and RolyPoly resulted in less significant trait-cell type associations compared to the MAGMA regression model (**Supplementary Fig. 7-8**)

and Supplementary Note 3).

- Changes in Supplementary Note 3
- Changes in Supplementary Fig. 7

Reviewer #2 (Remarks to the Author):

The authors mostly addressed my previous specific comments. One specific comment of mine which I felt the authors missed the point on was the relationship of their approach to approaches using single-cell ATAC-seq data to implicate specific cell types associated with GWAS. I was previously suggesting the authors should consider having some discussion on the relative merits of implicating cell-types directly from single cell ATAC-seq vs. using single cell gene expression data. Cusanovich et al, Cell 2018 showed that specific cell types could be implicated directly without needing to associate ATAC-seq signal with genes. I was not proposing, as the authors seem to suggest in their response of assigning ATAC-seq to specific genes to conduct enrichment analyses. For the problem that the authors motivate of identifying the relevant cell type for GWAS, that step does not seem necessary if one uses analysis approaches such as LDSC in conjunction with ATAC-seq data as shown in Cusanovich et al, Cell 2018.

Thanks for the clarification. We have now included a brief discussion about the use of ATAC-seq in the main text.

- Changes in the main text (from line 387)
We note that cell type specificity can be also implicated by other types of data resources such as ATAC-seq⁸⁴ and chromatin markers⁸⁵. We focused on scRNA-seq in this manuscript because of the relatively large availability of the datasets. In addition, the MAGMA gene-property analysis requires annotations per gene while ATAC-seq and chromatin markers are genome-based annotations, although this may not be a problem for approaches such as LDSC where annotations at SNPs level are used. When such data resources become available more abundantly in single cell resolution, they might also be advantageous for the identification of cell type specificity of a trait.

I was also asked to comment on the comments of Reviewer 3. I am not sure I fully understand the reviewer's previous point on the Skene's et al model. The authors have added a comparison with RolyPoly as well as stLDSC, which also addressed a comment I previously raised. My overall impression from the comparisons with these previous models is the authors model is doing something different and reasonable, but a ground truth is lacking to be able to make definitive statements of the relative sensitivity and specificity of different approaches.

In terms of the general issue of the significance of the biological results raised both in my review and of reviewer 3, I agree many of the observations were well established, though the authors do also point out some more novel/specialized results. I see the issue more being to what extent one can be confident about more novel and specialized results since they seem to be more dependent on the specifics on how the analysis is conducted. This issue is seen with the removal of the novel result of the relationship between pancreatic cells and brain-related

traits, highlighted by reviewer 1, between the original submission and this revised submission. Some of the results are also dependent on the specifics of which cell types are being conditioning on, something which the authors are more transparent about in the revised version.

More generally, the issue was raised both in my review and of reviewer 3 of the overall significance of this work. My overall impression of this manuscript is that it is a collection of incremental advances/observations (e.g. analyzing a large number of single-cell RNA-seq data, incorporating in a web-tool, the technical differences between the Skene et al and their model, the added workflow for conditional analyses), but I can also see the argument that collectively this work then becomes significant and will be useful to people.

We thank this reviewer for also commenting on reviewer #3's report and agree with this assessment.